# PSD95 as a New Potential Therapeutic Target of Osteoarthritis: A Study of the Identification of Hub Genes through Self-Contrast Model

**DOI:** 10.3390/ijms241914682

**Published:** 2023-09-28

**Authors:** Ping Huang, Jieming Lin, Hongxing Shen, Xiang Zhao

**Affiliations:** 1Department of Orthopaedics, Renji Hospital, School of Medicine, Shanghai Jiao Tong University, Shanghai 200127, China; huangping1480@renji.com (P.H.); 21432@renji.com (J.L.); 2Department of Spine Surgery, Renji Hospital, School of Medicine, Shanghai Jiao Tong University, Shanghai 200127, China

**Keywords:** osteoarthritis, RNA sequencing, hub gene, PSD95, *DLG4*

## Abstract

Osteoarthritis (OA) is a worldwide joint disease. However, the precise mechanism causing OA remains unclear. Our primary aim was to identify vital biomarkers associated with the mechano-inflammatory aspect of OA, providing potential diagnostic and therapeutic targets for OA. Thirty OA patients who underwent total knee arthroplasty were recruited, and cartilage samples were obtained from both the lateral tibial plateau (LTP) and medial tibial plateau (MTP). GO and KEGG enrichment analyses were performed, and the protein–protein interaction (PPI) assessment was conducted for hub genes. The effect of PSD95 inhibition on cartilage degeneration was also conducted and analyzed. A total of 1247 upregulated and 244 downregulated DEGs were identified. Significant differences were observed between MTP and LTP in mechanical stress-related genes and activated sensory neurons based on a self-contrast model of human knee OA. Cluster analysis identified *DLG4* as the hub gene. Cyclic loading stress increased PSD95 (encoded by *DLG4*) expression in LTP cartilage, and PSD95 inhibitors could alleviate OA progression. This study suggests that inhibiting PSD95 could be a potential therapeutic strategy for preventing articular cartilage degradation.

## 1. Introduction

Knee osteoarthritis (OA) is a widespread joint disorder, especially among older adults over 60 [1,2]. No practical methods are available to reverse cartilage degeneration [3]. The abnormal distribution of mechanical stress on articular cartilage and chronic low-grade inflammation is crucial in promoting the development of knee OA [4,5,6]. However, the molecular characteristics underlying the pathogenesis of osteoarthritis (OA) are still not fully understood [7]. Various omics analyses, such as genomics [8,9], transcriptomics [10], proteomics [11], and metabolomics, have been employed to gain insights into the complex mechanisms involved. These approaches collectively offer a comprehensive understanding of the molecular changes occurring in OA.

In previous studies, the approach of comparing healthy cartilage with osteoarthritis (OA) cartilage has been commonly employed to investigate the molecular characteristics associated with abnormal expression [9,12]. However, acquiring healthy cartilage samples raises ethical concerns and poses significant challenges. To overcome this persistent issue, we have proposed using a self-contrast model based on uneven cartilage degeneration observed in OA patients to comprehend human knee OA’s biological mechanisms better and identify potential therapeutic targets [13]. The model utilized relatively healthy cartilage in the lateral tibial plateau (LTP) as the control group and severely worn articular cartilage in the medial tibial plateau (MTP) as the degeneration group. This model reduces potential confounding factors, providing a valuable tool for investigating the biological causes of human knee OA and identifying therapeutic interventions [14]. The primary distinguishing factor between this model’s control and degeneration groups is the different mechanical stress experienced by cartilage in the LTP and MTP [15,16]. In OA, the cartilage in the MTP is exposed to high levels of mechanical overloading [17,18].

Despite the importance of the self-contrast model of human knee OA, there is a lack of high-throughput sequencing studies utilizing this model. Therefore, we performed RNA sequencing to identify the hub gene in OA, using the self-contrast model in this study. Our primary aim was to identify vital biomarkers and provide potential diagnostic and therapeutic targets for OA. Here, we identified a hub gene, *DLG4*, which was significantly upregulated in cartilage tissues from OA patients. Furthermore, we investigated whether inhibition of PSD95 (encoded by *DLG4*) could be a potential therapeutic strategy to prevent articular cartilage degradation.

## 2. Results

### 2.1. Genome-Wide Transcriptional Analysis of Cartilages Based on the Self-Contrast Model of Human Knee OA

This study was conducted from 16 December 2019 to 16 December 2022, and finally enrolled 10 patients for RNA sequencing. Cartilage samples were extracted from the lateral tibial plateau (LTP) and medial tibial plateau (MTP), as depicted in Figure 1A. Moreover, MTP and LTP samples from the same individual were self-contrasted, as shown in Figure 1B. The OARSI score was based on saffron O fast green staining and revealed severe degeneration in MTP cartilage samples, while relatively mild degeneration was observed in LTP cartilage samples (Figure 1C,D).

Genomic changes between the two groups were analyzed. The results showed that the batch effect was successfully eliminated (Figure 1E,F). In total, 1491 DEGs were identified between LTP and MTP, including 1247 upregulated genes and 244 downregulated genes. The resulting DEGs are shown as a volcano plot (Figure 1G) and a heatmap (Figure 1H). To elucidate the functional implications of the DEGs, we performed GO and KEGG enrichment analysis on the upregulated and downregulated DEGs. GO enrichment analysis showed that upregulated DEGs were mainly involved in the plasma membrane (GO:0005886; 200 DEGs) and integral component of the plasma membrane (GO:0005887; 79 DEGs), while downregulated DEGs were primarily associated with lamin binding (GO:0005521; 3 DEGs) and the regulation of the Wnt signaling pathway (GO:0030111; 3 DEGs). KEGG analysis revealed that upregulated DEGs were primarily enriched in pathways such as glutamatergic synapse (hsa04724; 14 DEGs) and ECM–receptor interaction (hsa04512; 9 DEGs), while downregulated DEGs were mainly enriched in pathways like protein digestion and absorption (hsa04974; 4 DEGs) and the ECM–receptor interaction (hsa04512; 3 DEGs).

GO and KEGG enrichment analyses of DEGs were performed, and an FDR < 0.05 was considered significantly enriched. We identified 25 biological processes (BPs), 17 molecular functions (MFs), and 19 cellular components (CCs) that were significantly overrepresented. The top 20 ranked BPs were involved in the urate metabolic process, cell–cell adhesion mediated by cadherin, the detection of external stimuli, and extracellular matrix organization (Figure 2A). The top 20 ranked CCs were involved in gamma-catenin binding, glutamate receptor activity, and Ras guanyl-nucleotide exchange factor activity (Figure 2B). The top 20 ranked MFs were associated with the microtubule-associated complex and transporter complex (Figure 2C). Finally, the top 20 ranked KEGG pathways were involved in histidine metabolism and rheumatoid arthritis (Figure 2D).

GSEA was performed to explore the biological behaviors of the MTP and LTP. MTP showed enriched hallmark pathways related to mechanical force, such as ECM receptor interaction, neuroactive ligand–receptor interaction, mineral absorption, calcium signaling pathway, and cytokine–cytokine receptor interaction, while olfactory transduction, glutamatergic synapse, and ribosome were enriched in LTP (Figure 2E,F). 

### 2.2. Mechanical Stimulus Signals and Activated Sensory Neurons Are Associated with the OA Pathogenesis Based on the Self-Contrast Model of Human Knee OA

To elucidate the functional implications of the DEGs, we performed pathway enrichment analysis via the application of Metascape to DEGs in cartilages from MTP versus LTP. The significantly overrepresented ontology terms were grouped into color-coded clusters based on their membership similarities and rendered as network plots. Twenty clusters of functions were overrepresented, encompassing the mechanical stimulus signal and activated sensory neurons, such as sensory perception of the mechanical stimulus, channel activity, and ion channel transport. Furthermore, some clusters were associated with collagen synthesis and degradation, such as extracellular matrix organization, cartilage development, and disassembly. Interestingly, the inflammation-related genes closely involved in the pathogenesis of OA failed to be enriched (Figure 3A). Therefore, the self-contrast model of human knee OA is a reliable model focusing on mechanically induced cartilage degeneration in terms of gene expression differences in MTP and LTP cartilages.

The enriched genes mainly included mechanosensitive ion channel activity and cellular response to mechanical stimulus. The expression level of mechanosensitive ion channel activity was downregulated, whereas other mechanical stress-related pathways were upregulated in MTP cartilage (Figure 4A and Appendix A). These findings suggest that mechanical stress-related pathways differ between MTP and LTP.

Activated neuron regulation, such as GRIN2A, DLG4, and NRXN1, was highly expressed in MTP cartilage. GO enrichment analysis shows that enriched genes were linked to the negative regulation of synapse organization (GO:0050808), axonogenesis (GO:0007409), and an ion channel complex (GO:0034702) (Figure 4B and Appendix A).

Given the diffuse inflammatory cytokines in the articular fluid, we also focused on the abnormal expression of inflammatory receptors, including IL-23R, IL-7R, and TNFRSF11B (Figure 4C and Appendix A). The high expression of these inflammatory receptors may lead to the excessive activation of inflammatory pathways in MTP cartilage. GO pathway enrichment analysis: enriched genes were linked to inflammatory factors, including cellular response to interleukin-6 (GO:0071354), and activation of NF-κB-inducing kinase activity (GO:0007250) (Figure 4C and Appendix A).

GO pathway enrichment analysis: enriched genes were linked to anti-inflammatory factors, including interleukin-10 secretion (GO:0032653) and cellular response to interleukin-4 (GO:0071353) (Figure 4D and Appendix A). Compared with LTP, MTP showed the activation of the anti-inflammatory cytokine pathway.

Glycosaminoglycan and collagen metabolic disorders presented typical clinical symptoms in OA [9,19]. Heatmaps show the elevated expression of osteogenesis differentiation in MTP, such as *SHH* and *RUNX3* (Appendix A). GO enrichment analysis shows enriched genes linked to positive regulation of osteoblast differentiation (GO:0045669), chondrocyte proliferation (GO:0035988) and negative regulation of chondrocyte differentiation (GO:0032331) (Figure 4E and Appendix A). 

Chronic low-grade inflammation promotes the consumption of NAD^+^ in OA, which inhibits the activity of Sirt1 and promotes the senescence of knee cartilage [20]. Heatmaps show reduced NAD^+^ biosynthesis enzyme NAMPT1 and NAMPT2 expression in MTP (Appendix A). GO enrichment analysis shows enriched genes were linked to replicative senescence (GO:0090399) and cellular senescence (GO:0090398) (Appendix A).

### 2.3. DLG4 Acts as a Hub Gene in Knee OA

The potential interactions of DRGs were visualized using Cytoscape (Figure 5A). We randomly selected 8 algorithms to calculate the score of the top 25 genes. Finally, two hub genes (*DLG4*, *MKi67*) were obtained and shown in Figure 5B.

Based on transcriptome data, we analyzed the relationship between OA and immune cell populations using microenvironment cell population counters. The results showed that B lineage was mainly associated with OA (r = 0.92) (Figure 6A,B). The abundance of activated CD4 T cells, mast cells, memory B cells, and central memory CD8 T cells was significantly higher in the MTP cartilages (Figure 6C). The correlations between them were calculated using Pearson’s correlation method and were visualized with a heat map (Figure 6D). 

### 2.4. Cyclic Loading Stress Increased DLG4 Expression in LTP Cartilage

After cyclic loading treatment (Figure 7A), the results of saffron O staining indicated that the LTP surface cartilage was worn (Figure 7B), and that the expression of *DLG4* was increased in the LTP cartilage samples after loading stress treated (Figure 7C). qRT-PCR and WB also showed that *DLG4* was upregulated in LTP cartilage after cyclic loading treatment (Figure 7C,D). Cyclic loading stress stimulation promoted the upregulation of inflammatory factors (Figure 7E–H).

### 2.5. PSD95 (DLG4 Encoded) Inhibitor Alleviates OA Progression

Immunofluorescence showed that the expression of PSD95 was significantly upregulated in the MTP cartilage based on the self-contrast model of human knee OA (Figure 8A). The expression of *DLG4* was significantly higher in MTP cartilage when compared to the LTP cartilage, which was in accordance with the results of our analysis (Figure 8B,C). These results suggest that PSD95 (*DLG4*) could be key biomarker for OA based on the self-contrast model of human knee OA. We used PSD95 (*DLG4*) inhibitor nerinetide to treat the cells from the MTP. Our results revealed that adding nerinetide increased the expression of COL2A1 (Figure 8D) and decreased the expression of MMP13, IL-1β, and IL-6 (Figure 8E,F). 

Compared to the sham-treated rats, the ACLT-treated rats displayed significant cartilage degeneration and subchondral bone remodeling (Figure 9A), which are typical features of OA. However, the rats treated with PSD95i showed significant improvement in these pathological features (Figure 9B), suggesting a potential protective effect of PSD95 inhibition on OA progression. Immunohistochemistry-based staining of p21 in the articular cartilage further supported these findings (Figure 9C), showing similar results. In summary, our study indicates that the pharmacological inhibition of PSD95 can significantly ameliorate the progression of OA (Figure 9D). 

## 3. Discussion

This study is the first to perform bioinformatics analysis based on a self-contrast model of human knee OA. Transcriptome analysis was performed to identify and validate the role of PSD95, encoded by the *DLG4* gene, in OA development. Inhibiting PSD95 alleviates OA symptoms, suggesting its potential as a therapeutic target. 

Another innovation of this study is focusing on the significance of mechanical stress and activated neuron regulation in OA, which differed from previous research that primarily focused on inflammation and cartilage degradation [12,21,22]. Recent studies have categorized OA into four subtypes [12]. Using a self-contrast design in our study, as opposed to previous studies with samples from healthy volunteers and OA patients, may have contributed to the differences in research outcomes. Our sequencing results identified five key features of OA: abnormal activation of mechanical stress signals, activation of neuronal ion channels, activation of inflammatory pathways, cellular senescence, and abnormal collagen metabolism in cartilage. 

This study also demonstrates the underlying biological mechanisms linking mechanical stress with osteoarthritis. Excessive mechanical stress on articular cartilage disrupts its structural integrity and impairs its ability to withstand mechanical forces [23,24]. This mechanical overload triggers a cascade of biochemical and cellular events [25,26], including the release of pro-inflammatory cytokines, matrix metalloproteinases, and other inflammatory mediators [27,28]. These factors further exacerbate cartilage damage and contribute to the chronic low-grade inflammation observed in knee OA. In OA, the increased pro-inflammatory factors can amplify the inflammatory response [29]. Chronic low-grade inflammation may result in the upregulation of cartilage catabolic factors and articular cartilage degradation [30]. For example, elevated expression of catabolic ADAMTS enzymes in OA promotes the degradation of the cartilage matrix [30]. Furthermore, chronic low-grade inflammation in OA has accelerated cellular senescence [31,32]. Increased production of pro-inflammatory mediators is a feature of the senescence-associated secretory phenotype (SASP), also described in OA senescence chondrocytes.

Our study identified the pivotal role of PSD95, encoded by the *DLG4* gene, in OA. We validated the abnormal overexpression of PSD95 in MTP cartilage using clinical samples. PSD95 is a protein associated with neuronal activation and is critical in neural transmission [33,34]. An alternative model similarly corroborated the role of PSD95 inhibition in osteoarthritis on the premise that cyclic loading stress promotes the expression of PSD95 in cartilage. The evidence from our research indicates that inhibiting PSD95 activity reduces the release of inflammatory factors in human OA chondrocytes and mitigates the progression of OA in animal models. These findings show the crucial role of the *DLG4* gene in the mechanical–inflammatory pathway and propose that modulating *DLG4* activity could offer therapeutic opportunities in managing the progression of osteoarthritis.

PSD95 forms the NOS-PSD95-NMDA receptor complex, mediating Ca^2+^ influx in response to mechanical stimulation [35]. NOS, which is increased in human osteoarthritis-affected cartilage but not in normal cartilage [36,37], mediates the effects of pro-inflammatory cytokines and induces chondrocyte apoptosis. NOS plays a role in the pathophysiology of OA [36], as evidenced by NOS inhibition restoring impaired ATP production, reducing inflammation, and delaying OA progression [38]. The activation of PSD95 may contribute to the synthesis and release of inflammatory mediators in OA via the NOS-NO pathway [39], thereby contributing to the establishment of a chronic inflammatory microenvironment in OA [40]. However, further investigation is required to fully understand the specific mechanisms by which PSD95 functions in OA. 

This study has certain limitations. (1) The number of included samples for RNA sequencing in OA cartilage is limited. (2) The underlying mechanisms by which PSD95 promotes OA were not extensively explored. Previous studies have reported that PSD95 can activate the NOS-NO pathways, suggesting its potential involvement in the release of inflammatory factors.

## 4. Materials and Methods

### 4.1. Study Design

We collected cartilage samples from 30 knee OA patients aged 55–65 undergoing total knee arthroplasty in Ren Ji Hospital. This study aims to provide observational research by using RNA sequencing. This study was conducted from 16 December 2019 to 16 December 2022. Patients with previous knee infections or rheumatic arthropathies were excluded from the study. Samples were obtained from the weight-bearing regions of the lateral tibial plateau (LTP) and the medial tibial plateau (MTP) [14]. Some MTP samples had insufficient cartilage or failed to extract cDNA, resulting in their exclusion from the study group. Eventually, only 10 OA samples obtained cartilage RNA from MTP and LTP. The participants were provided with written information about the study and must consent in writing before participating. All human experiments in this study adhered to the regulations of the Ethics Committee at Ren Ji Hospital (ethical approval identification number: 2019-166, approved on 16 December 2019).

### 4.2. Histological Analysis

The knee specimens were fixed in 4% PFA for 25 h and decalcified with a 10% ethylenediaminetetraacetic acid (EDTA) solution for 4 weeks. After the completion of decalcification, the samples were embedded in paraffin blocks. Coronal sections of the knee were consecutively taken for saffron-O/fast green staining. The severity of articular cartilage injury was assessed using the OARSI scoring system. 

### 4.3. Data Processing and Assembly

Data processing and assembly Raw reads underwent quality filtering to remove reads with low quality and adapter contamination using TrimGalore (v0.6.4). The clean reads were mapped back to this reference using Bowtie (v1.2.3). Unigene counts were then quantified using RSEM (v1.3.3). Data were normalized using the TPM method to account for differences in sequencing depth [41].

### 4.4. Identification of Differential Expression Genes and Enrichment Analysis

DEseq2 R package (v1.28.1) was used for differential expression analysis with raw read counts [42]. Adjusted *p* values were obtained using the Benjamini–Hochberg method. DEGs were selected with adjusted *p* values < 0.05 and |log2 (fold change)| > 2. The cluster Profiler R Package (v3.16.1) was utilized for GO and KEGG enrichment analysis [43]. 

### 4.5. Protein–Protein Interaction (PPI) Network Construction

PPI network construction Enrichment analysis was performed by all DEGs as the background [42]. Metascape was utilized to identify significantly overrepresented ontology terms (*p* < 0.01) with a minimum count of 3 and an enrichment factor greater than 1.5. These terms were then grouped into clusters based on their similarity and visualized as a network plot, where terms with a similarity greater than 0.3 were connected.

The STRING database explored potential interactions among the DEGs and constructed PPI networks. PPI pairs were extracted based on a medium confidence threshold of >0.4. Cytoscape (version 3.9.1) was used for network visualization, and the cytoHubba plugin was utilized to calculate the ranking of DEGs. The top 25 genes according to eight algorithms (MCC, MNC, Degree, BottleNeck, EcCentricity, Closeness, Radiality, Stress) in cytoHubba were selected. Finally, hub genes closely associated with the development of osteoarthritis were identified using the R package “UpSetR”.

### 4.6. Assessment of Osteoarthritis-Related Immune Infiltration

The immune microenvironment encompasses various components, including immune cells, inflammatory cells, fibroblasts, and cytokines. Analysis of immune cell infiltration plays a crucial role in predicting disease progression. The GSEA algorithm assessed the abundance of 28 immune cell types. The “corrplot” package was utilized to calculate the Spearman rank correlation coefficient.

### 4.7. Cartilage Cyclic Loading Model

The MTP cartilage (from OA patients, obtain ethical approval) was divided into approximately 8 × 8 mm pieces wrapped in bilateral sterile gloves (ethical approval identification number: 2019-166). Following a 380–600 N cyclic load, samples were loaded between −5.5 ± 0.2 MPa and −9.3 ± 0.2 MPa stress. Samples were loaded to 3600 cycles at a frequency of 1 Hz (*n* = 3). Then, samples were cultured in a culture medium (same as chondrocyte medium) for six days in vitro.

### 4.8. Isolation of Chondrocytes from MTP and LTP

Articular cartilage from MTP and LTP from OA cartilage were subjected to continuous collagenase digestion to isolate cartilage cells, respectively. Chondrocytes were maintained in DMEM/F12 supplemented with 10% FBS (Gibco, Grand Island, NY, USA; lot number 1652790). Chondrocytes were grown in standard conditions. We used primary and the first passage chondrocytes.

### 4.9. qRT-PCR

Total RNA from human samples or cells was extracted using the TRIzol reagent (Invitrogen) and quantified by a NanoDrop spectrophotometer. Complementary cDNA was synthesized from total RNA using a 5X All-In-One MasterMix kit (ABM). This system utilized SYBR Green dye to detect and quantify the amplification of specific target genes. The obtained data were normalized to GAPDH to determine relative gene expression levels, which serves as an internal control. The fold change between samples or experimental conditions was calculated by the 2^−∆∆Ct^ method. Primer sequences used in our study are shown in Appendix A.

### 4.10. Animals and Experiments

Nine 10-week-old female SD rats were randomly assigned to three groups (*n* = 3): Sham, ACLT, and ACLT + PSD95 (*DLG4*)i (Tat-NR2B9c). Anesthesia for the study was induced via the intraperitoneal injection of pentobarbital sodium at 35 mg/kg. Anterior cruciate ligament transection (ACLT) surgery was performed to generate an OA mouse model by transecting ACL. The sham group underwent a similar procedure without exposing the articular capsule. The ACLT + PSD95 (*DLG4*)I group received oral gavage of Tat-NR2B9c once daily for four weeks, while the sham and ACLT groups received normal saline treatment. After four weeks, the rats were euthanized using 3% isoflurane. Knee micro-CT examinations were conducted. Knee joints were fixed in 4% PFA for two days at room temperature. The procedures involved in the study were approved by the Scientific Research Ethics Committee of Zhengzhou Weisa Biotechnology Co., Ltd. (No. V3A02022109012).

### 4.11. Statistical Analysis

Data were analyzed using Prism (v 5.0b, GraphPad Software). Data were analyzed using Prism (v 5.0b, GraphPad Software). Quantitative results were reported as means ±  SD. The Kolmogorov–Smirnov and Shapiro–Wilk tests were performed to determine the data distribution. For normally distributed data, an unpaired *t* test or unpaired *t* test with Welch’s correction (two groups) and one-way ANOVA (multiple groups) were used for analysis. *p* < 0.05 was considered statistically significant.

## 5. Conclusions

Our sequencing analysis of a human knee OA self-contrast model revealed two main characteristics of human knee OA cartilage: abnormal activation of mechanical stress signals and activated neuron regulation. *DLG4*, identified as the hub gene in OA, could be a potential target for alleviating OA. 

## Figures and Tables

**Figure 1 ijms-24-14682-f001:**
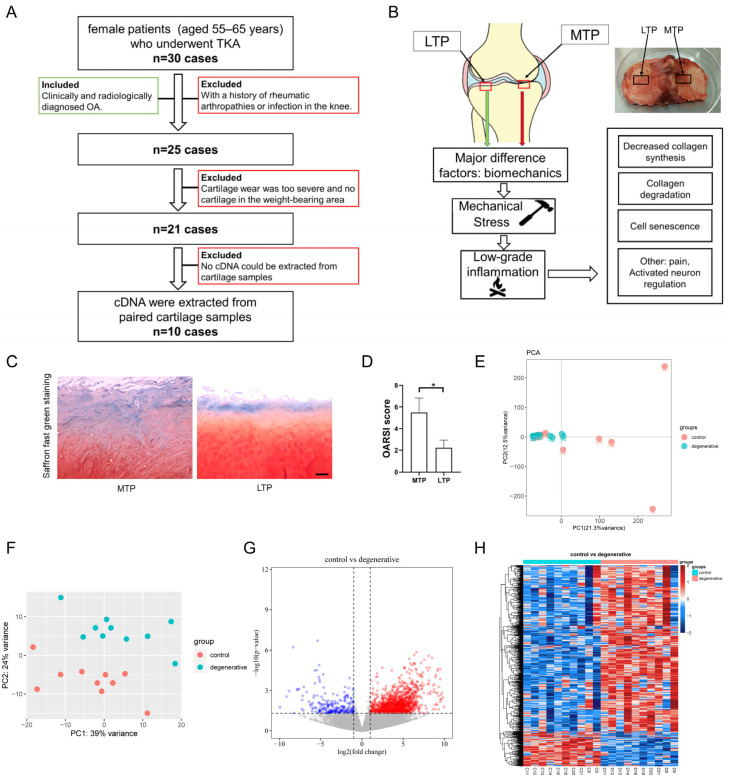
The panoramic comparison is based on the self-contrast model of human knee OA and the identification of DEGs. (**A**) Flowchart of human OA knee cartilage sample collection. (**B**) Schematic diagram of a mechano-inflammatory mode of action in the presence of articular cartilage in OA. (**C**) MTP and LTP cartilage samples were stained with saffron O fast green staining. Scale bar: 100 μm. (**D**) OARSI score based on saffron O fast green staining evaluated severe degeneration in MTP and LTP cartilage samples. * *p* < 0.05. (**E**,**F**) PCA analyses were performed to visualize the effect before (**E**) and after (**F**)**,** removing the batch effect. (**G**) A volcano plot was generated to depict the differentially expressed genes (DEGs; fold change >2 or <−2, aducsted *p*-value < 0.05) between MTP and LTP. Upregulated genes are marked in light red; downregulated genes are marked in light green. (**H**) Heatmap with unsupervised hierarchical clustering showing the expression profile of DEGs in human cartilage. Each row represents one sample, and each column represents one DEG. The scaled expression value (column z-score) is displayed in a blue–red color scheme, with blue and red indicating low and high expression, respectively.

**Figure 2 ijms-24-14682-f002:**
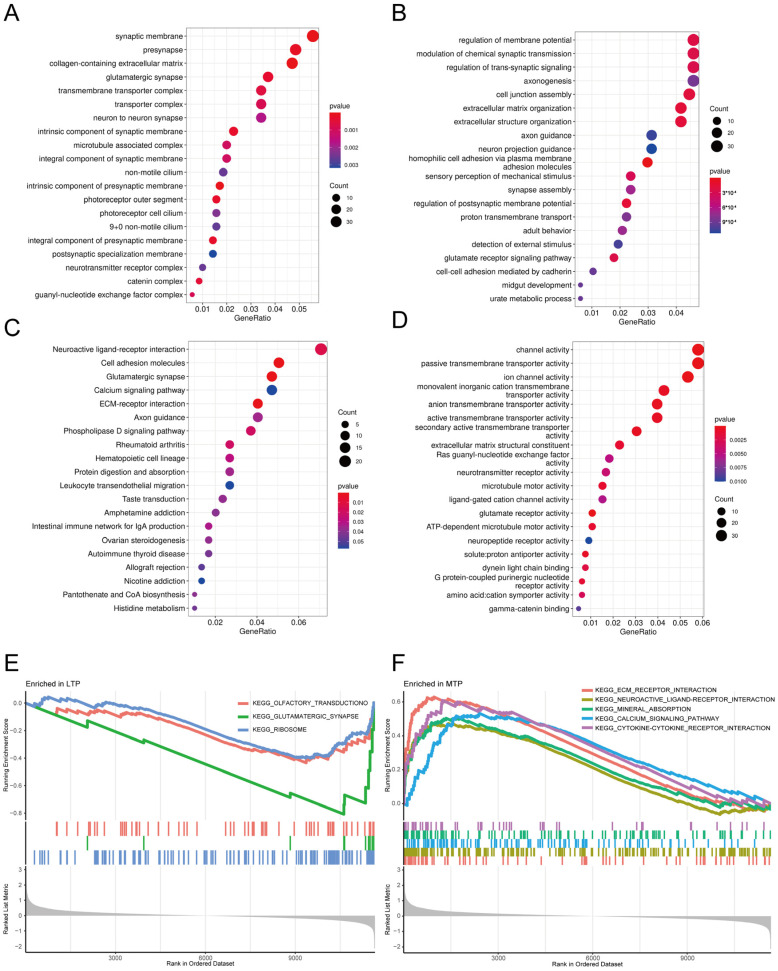
Functional enrichment analysis of DEGs. (**A**–**D**) Bubble charts show GO-enriched items of DEGs in three functional groups: biological processes (BP, (**A**)), cell composition (CC, (**B**)), molecular functions (MF, (**C**)), and KEGG-enriched items (**D**). The *x* axis labels represent gene ratios, and the *y* axis labels represent GO or KEGG terms. Different colors of circles represent different adjusted *p* values. (**E**,**F**) GSEA analysis of the Reactome pathway includes LTP (**E**) and MTP (**F**) of the top pathways.

**Figure 3 ijms-24-14682-f003:**
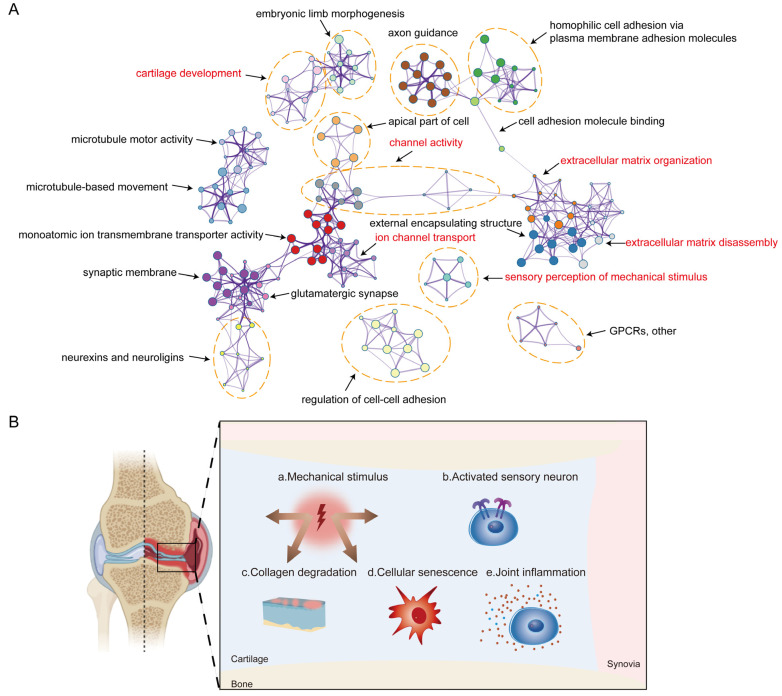
Transcriptome changes between MTP and LTP. (**A**) Metascape-based functional enrichment analysis on DEGs in cartilage from MTP versus LTP. (**B**) Schematic review of the five major characteristics of knee osteoarthritis based on the self-contrast model of human knee OA.

**Figure 4 ijms-24-14682-f004:**
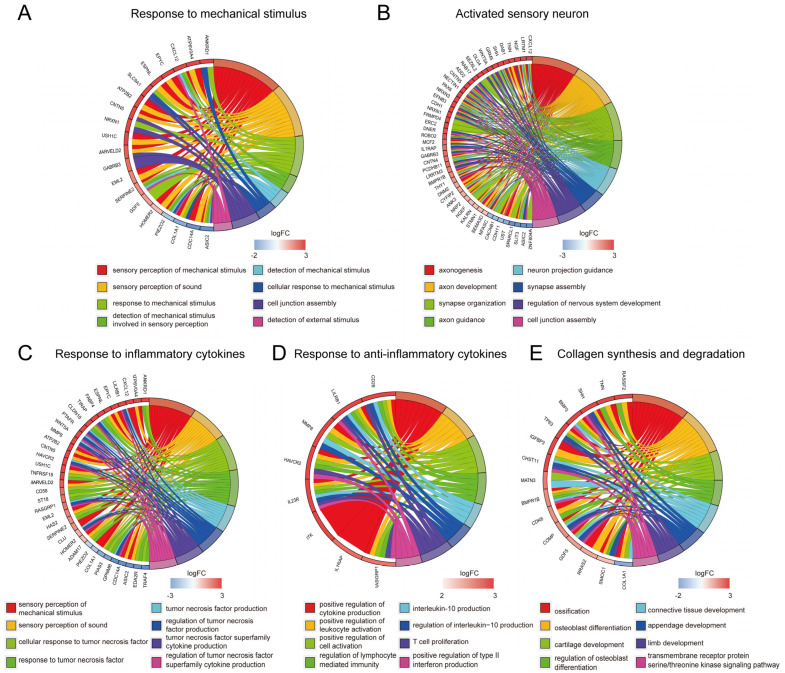
Gene-level characterization of MTP and LTP cartilage transcriptome changes based on the five major characteristics of knee OA. (**A**–**E**) Chord plots show GO-enriched items of DEGs. The graph presents symbols of differentially expressed genes (DEGs) on the left side, with their corresponding fold change values represented by a color scale. Colored connecting lines indicate the involvement of these genes in gene ontology (GO) terms.

**Figure 5 ijms-24-14682-f005:**
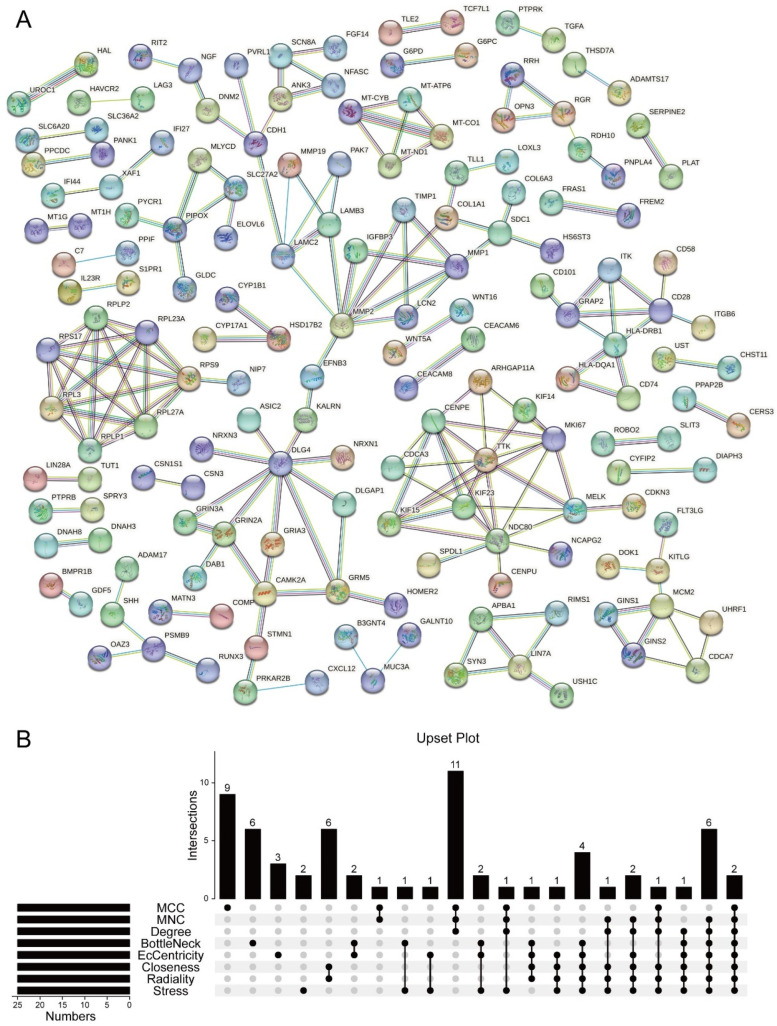
Construction of the PPI network and screening of key genes. (**A**) Differentially expressed genes PPI network constructed using STRING database. (**B**) The Venn diagram showed that eight algorithms screen out 2 overlapping hub genes.

**Figure 6 ijms-24-14682-f006:**
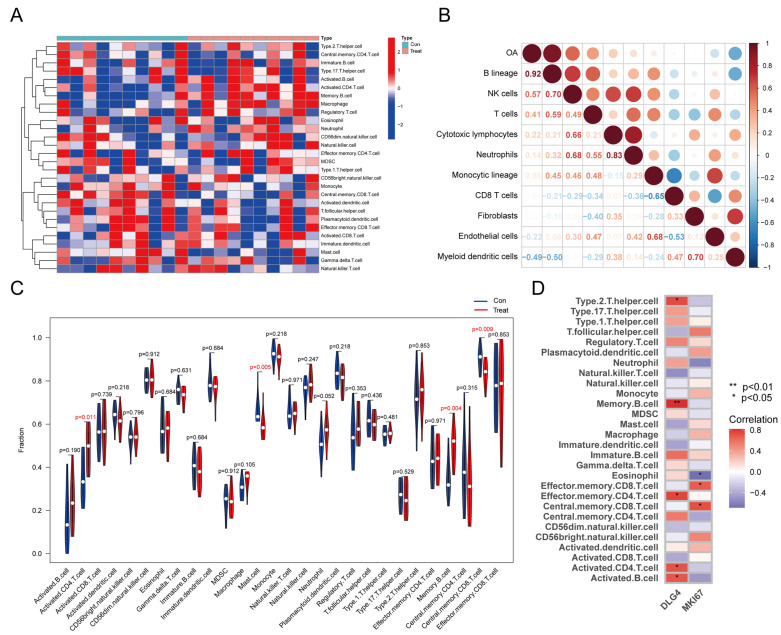
Analysis of the immune infiltrating cells between MTP and LTP cartilages. (**A**) A heatmap was generated to visualize the relative abundance of immune infiltrating cells between MTP and LTP cartilages. (**B**) Correlation analysis between OA and immune infiltration. (**C**) Violin plot showing the fraction of immune cell infiltration by typing in OA samples based on the self-contrast model. Blue indicates LTP, and red indicates MTP. (**D**) Correlations between hug genes *DLG4*, *Mik67*, and infiltrating immune cells.

**Figure 7 ijms-24-14682-f007:**
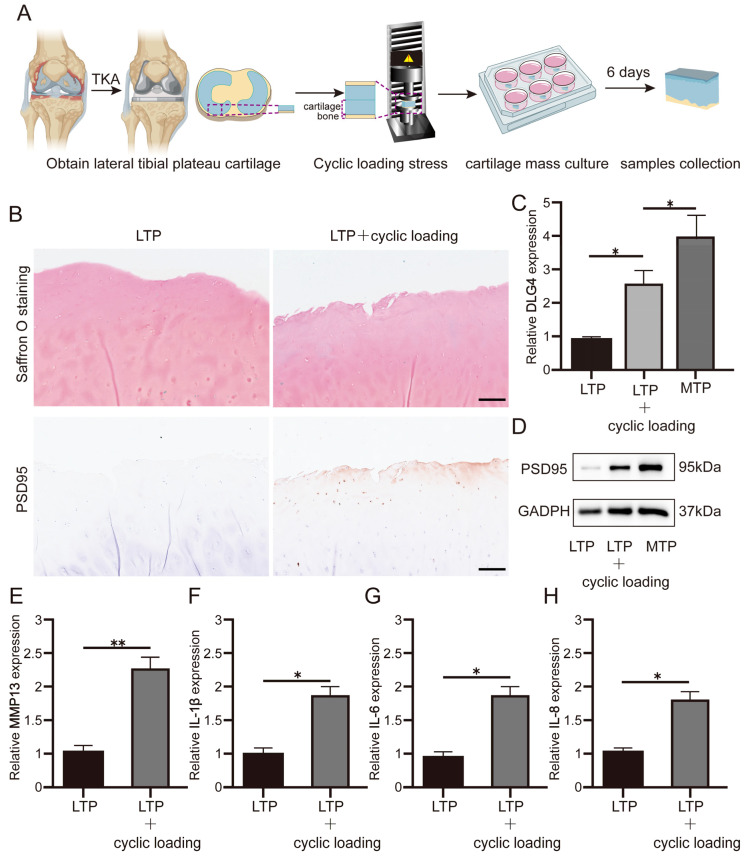
Cyclic loading stress increased PSD95 expression in cartilage. (**A**) Cyclic stress loading instrument for LTP cartilage. (**B**–**D**) Tissue damage was evaluated by saffron O staining, and PSD95 and *DLG4* expression was evaluated by immunohistochemistry. Scale bar: 100μm. (**E**–**H**) RT-PCR detected the expression of MMP13, IL-1β, IL-6, and IL-8. Data are the mean ± SD 3 independent experiments. * *p* < 0.05, ** *p* < 0.01. Scale bar, 100 μm.

**Figure 8 ijms-24-14682-f008:**
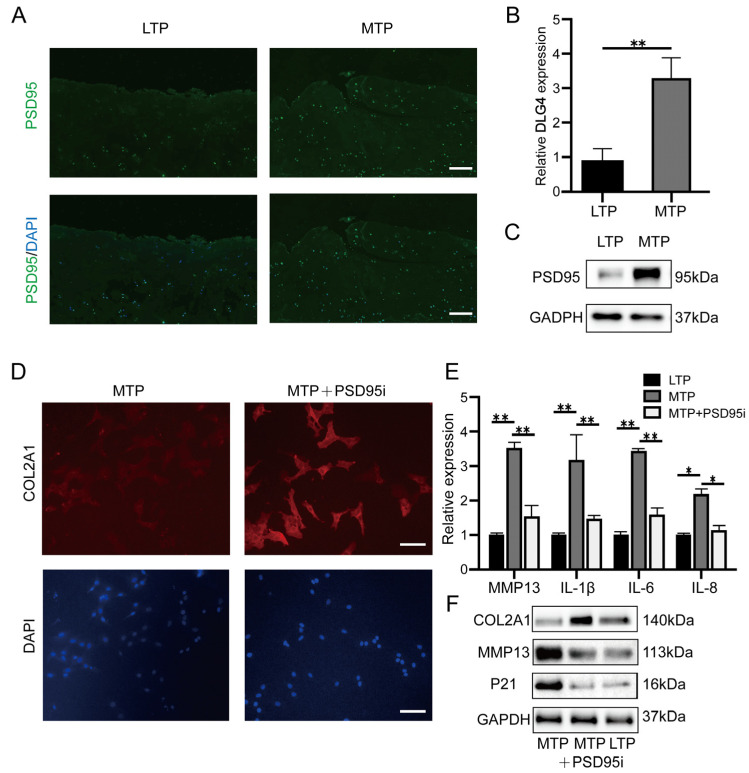
Validation PSD95 inhibitors promoted collagen synthesis and reduced the secretion of inflammatory factors in MTP cartilage. (**A**) Immunostaining for PSD95 in MTP and LTP cartilage from the same human knee OA specimen. (**B**) Validation of differential expression of *DLG4* in LTP and MTP samples by qRT-PCR (*n* = 6). (**C**) PSD95 protein expression in MTP and LTP samples by Western blotting. (**D**) Immunostaining for COL2A1 in chondrocytes from the MTP cartilage treated with PSD95 inhibitors. (**E**,**F**) PSD95 inhibitors decreased the MMP13 and p21 expression and increased the COL2A1 expression in MTP chondrocytes. Data are the mean ± SD 3 independent experiments. * *p* < 0.05, ** *p* < 0.01. Scale bar, 100 μm.

**Figure 9 ijms-24-14682-f009:**
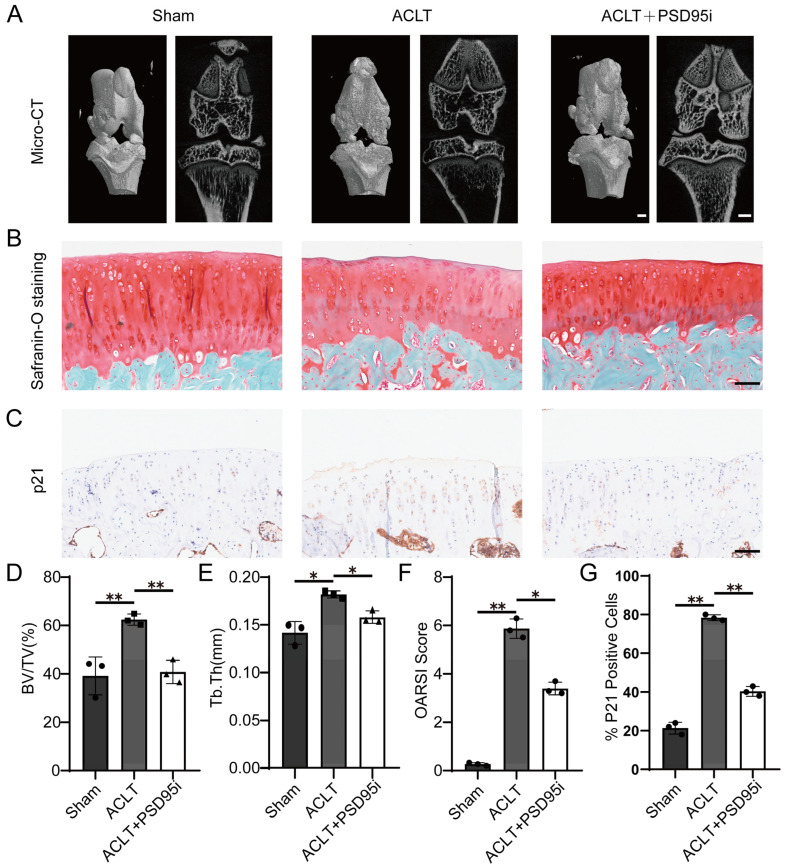
PSD95’s pharmacological inhibition attenuates cartilage degeneration and chondrocyte senescence. (**A**) Representative micro-CT images from medial femorotibial joints of SD rats 4 weeks after sham surgery or ACLT or ACLT + PSD95i. Scale bar: 1000 μm. (**B**) Representative images of Safranin O staining of cartilage tissue from medial femorotibial joints of SD rats 4 weeks after sham surgery or ACLT or ACLT + PSD95i. Scale bar: 100 μm. (**C**) Representative images of immunohistochemistry of p21^CIP1^ in cartilage tissue from medial femorotibial joints of SD rats 4 weeks after sham surgery or ACLT or ACLT + PSD95i. Scale bar: 100 μm. (**D**,**E**) Quantitative analysis of structural parameters of knee joint subchondral bone: bone volume/tissue volume (BV/TV, %) (**D**), and trabecular thickness (Tb.Th, mm) (**E**). (**F**) Calculation of Osteoarthritis Research Society International (OARSI) scores. (**G**) Quantification of the number of p21^+^ cells per field of view. The results are presented as the mean ± SD, n = 3 mice per group, * *p* < 0.05, ** *p* < 0.01.

## Data Availability

To protect the biological information and privacy of the donors of this study, the raw data are not to be shared publicly but are available on request from the authors.

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
