# Peer review of "PSD95 as a New Potential Therapeutic Target of Osteoarthritis: A Study of the Identification of Hub Genes through Self-Contrast Model"

_ijms, 2023, doi:10.3390/ijms241914682_

Round 1

Reviewer 1 Report

On the abstract

1. Line 22: "Cluster analysis revealed that the hub gene DLG4 was obtained." could be sounding better as "Cluster analysis identified DLG4 as the hub gene."

On the manuscript

1. There should be a space before the reference, i.e., "degeneration [2]." There are many occurrences like this.

2. Row  54 - "different" should be used instead of "differential"

3. Concepts regarding the effects of mechanical stress, inflammation, and cartilage degradation are reiterated multiple times throughout the introduction - the authors should combine these concepts and forge different hypotheses instead.

4. It would be helpful to lead into it with more context about why existing models or studies were insufficient

5. Authors mention that the primary distinguishing factor between the control and degeneration groups is the mechanical stress - prior information about inflammation and cellular changes makes it seem as though mechanical stress is only a part of the picture, not the sole differentiator. Please clarify.

6. In row 345  it should be "ligament transection"

7. The method starts by saying they collected samples from 30 patients, but eventually, only 10 OA samples were analyzed - how and why the other samples were excluded?

8. after euthanizing the rats, it says they were "submitted to cervical dislocation." This is redundant and might be conceptually misleading. If euthanasia was carried out with isoflurane, there might not be a need for cervical dislocation?

9. the reader might wonder if the samples were taken from sites of the knee with OA or from other areas. More clarity on the specific location and severity of OA

Author Response

Responses to the Referee #1:

On the abstract

  1. Line 22: "Cluster analysis revealed that the hub gene DLG4 was obtained." could be sounding better as "Cluster analysis identified DLG4 as the hub gene."

Response: We thank you for your suggestions.

We have modified the description to 'Cluster analysis identified DLG4 as the hub gene' on line 23 of page 1.

On the manuscript

  1. There should be a space before the reference, i.e., "degeneration [2]." There are many occurrences like this.

Response: We thank the reviewer for his/her helpful suggestions. We have revised the manuscript and included a space before each reference.

  1. Row 54 - "different" should be used instead of "differential"

Response: According to the reviewer's suggestions,

we have modified the description to ' different ' on line 65,page 2.

  1. Concepts regarding the effects of mechanical stress, inflammation, and cartilage degradation are reiterated multiple times throughout the introduction - the authors should combine these concepts and forge different hypotheses instead.

Response: We thank the reviewer for the thoughtful comments.

We incorporated these concepts (regarding the effects of mechanical stress, inflammation, and cartilage degradation) into the discussion section to integrate the sequencing results suggesting the activation of mechanical stress pathways, inflammation-related pathways, and the consequences of cartilage degeneration. This integration was done to elucidate the potential direct link between excessive mechanical stress, inflammation, and degeneration. In the revised manuscript, the corresponding discussion has been supplemented in lines 277-290 of page 15.

  1. It would be helpful to lead into it with more context about why existing models or studies were insufficient instead.

Response: We thank the reviewer for the kind advice.

To elucidate the advantages of the model used in our study, we introduce the main approaches for establishing control groups in previous studies. “In previous studies, the approach of comparing healthy cartilage with osteoarthritis (OA) cartilage has been commonly employed to investigate the molecular characteristics associated with abnormal expression. However, acquiring healthy cartilage samples raises ethical concerns and poses significant challenges.” The corresponding revisions have been added to lines 41-45, page 1 of the revised manuscript.

  1. Authors mention that the primary distinguishing factor between the control and degeneration groups is the mechanical stress - prior information about inflammation and cellular changes makes it seem as though mechanical stress is only a part of the picture, not the sole differentiator. Please clarify.

Response: We thank the reviewer for his/her thoughtful consideration.

Based on our team's preliminary research,  which utilized finite element analysis,we identified an uneven distribution of stress on the degenerative cartilage surface in individuals with osteoarthritis (OA). As OA progressed, early-stage patients showed increased mechanical stress peaks and affected areas on the medial tibial plateau (MTP) cartilage surface. In severe cases of OA, the stress peaks on the MTP cartilage surface were even more significant. Additionally, inflammatory mediators and immune cells were diffusely present in the knee joint cavity, with almost equal opportunities for contact with both MTP and lateral tibial plateau (LTP) cartilage. Due to MTP cartilage wear in patients with OA, there will be varus deformity, which leads to internal displacement of force lines and aggravates cartilage injury. That is, the weight bearing in MTP cartilage area is increased. To solve this problem, Chinese scholars perform high tibial osteotomy(HTO) for OA patients. The main principle is to change the biological force line and reduce the weight bearing of MTP cartilage.

In OA, the cartilage in the MTP is exposed to excessive mechanical overloading compared to LTP cartilage. This excessive mechanical stress accelerates cartilage degeneration and is the primary difference factor for the imbalanced degeneration observed in the medial and lateral compartments. Unlike previous studies that used healthy controls, the self-contrast design employed in this research allows for a more focused investigation into the mechanisms underlying cartilage degeneration associated with mechanical stress.

  1. In row 345  it should be "ligament transection"

Response: We thank you for your careful examination.

We have modified the description to ' ligament transection '.

  1. The method starts by saying they collected samples from 30 patients, but eventually, only 10 OA samples were analyzed - how and why the other samples were excluded?

Response: We thank you for your suggestions.

Due to limitations in patient availabilityt, we analyzed the data from 10 out of 30 patients in this manuscript. The main reason was that the amount of MTP cartilage was very few, and the content of chondrocytes was small. Therefore, the MTP cartilage of some samples might not be able to lift cDNA, leading to the exclusion criteria. Another reason is some of the included samples of degenerative cartilage had no cartilage tissue. Overall, to ensure the validity of the data and the reliability of our conclusions, we only analyzed samples from effective tissues. In the Results section of the manuscript, we have made the following modifications: “This study was conducted from December 16, 2019, to December 16, 2022, and finally enrolled 10 patients for RNA sequencing.” (line 79-80, page 2)

  1. after euthanizing the rats, it says they were "submitted to cervical dislocation." This is redundant and might be conceptually misleading. If euthanasia was carried out with isoflurane, there might not be a need for cervical dislocation?

Response: We thank you for your careful examination

In the method, we have deleted this paragraph.

  1. the reader might wonder if the samples were taken from sites of the knee with OA or from other areas. More clarity on the specific location and severity of OA.

Response: We thank you for your suggestions.

In order to eliminate any confusion for the readers and to make the manuscript's description clearer, we have added visual photographs of OA tibial plateau samples obtained through TKA surgery, along with labeled indications of the site for cartilage sampling in figure 1B.

Reviewer 2 Report

This study entitled “PSD95 as a new potential therapeutic target of osteoarthritis: identified by hub genes through the self-contrast model” seems to have been generally well executed and written. Furthermore, I believe that this paper will be of great interest to the readers. However, I have some remarks that require authors attention.

Title

Please add the type of article in your title.

Abstract

Please make subsections in the Abstract, i.e., Background/Methods/Results/Conclusion.

Keywords

Consider some additional MeSH keywords to improve the visibility of your article among readers.

1. Introduction

I suggest you to expand your Introduction with the recent relevant evidence, such as the study of Caric et al, Int J Mol Sci, 2021.

Please state the clear hypothesis of your study at the end of Introduction.

4. Materials and methods

I recommend you to named the first subsection Study design, and begin this section with an information what type of study you have performed, in which time period and where. This subsection should contain the Ethical approval for conducting the study which you have already stated, however please state the date when the Ethical approval was gained. If you register your study on any protocol this information should also be stated here (e.g., ClinicalTrials.gov).

4.10. Statistical analysis

Should be for some results a more appropriate to show them as median and IQR and then analyze by Wilcoxon test?

How did you arrive to the final sample size. Did you perform any type of sample size calculation?

2. Results

Please begin this section with the period of conducting your study and with the information of the number if patients ultimately analyzed.

3. Discussion

Discussion section is definitely too short, so try to better explain your Results and compare them and further expand with the findings of recent similar studies.

Please state the limitations of your study at the end of Discussion.

References

Please add some relevant recent references, firstly to expand the Discussion section of your work.

Author Response

Responses to the Referee #2:

This study entitled “PSD95 as a new potential therapeutic target of osteoarthritis: identified by hub genes through the self-contrast model” seems to have been generally well executed and written. Furthermore, I believe that this paper will be of great interest to the readers. However, I have some remarks that require authors attention.

Title

Please add the type of article in your title.

Response: We thank you for your suggestions.

Our study focuses on the identification of hub genes using a self-contrast model, specifically highlighting PSD95 as a promising therapeutic target for osteoarthritis. To distinguish it from the review, we changed the title to ‘PSD95 as a new potential therapeutic target of osteoarthritis: A study of the identification of hub genes through self-contrast model’.

Abstract

Please make subsections in the Abstract, i.e., Background/Methods/Results/Conclusion.

Response: We thank you for your kind suggestions.

The journal of IJMS may not need to write the abstract with subsections. To comply with the journal's formatting requirements, we carefully reviewed the manuscript throughout.

Keywords

Consider some additional MeSH keywords to improve the visibility of your article among readers.

Response: We thank the reviewer for the kind advice.

We have added and finally confirmed the following as keywords: Osteoarthritis, RNA sequencing, hub gene, PSD95, DLG4.

  1. Introduction

I suggest you to expand your Introduction with the recent relevant evidence, such as the study of Caric et al, Int J Mol Sci, 2021.

Please state the clear hypothesis of your study at the end of Introduction.

Response: We thank you for your suggestions.

We have added a hypothesis at the end of the Introduction (line 72-75, page 2). We have made modifications to the introduction section to provide a more concise explanation of our methodology for establishing a control group, the distinctions from prior studies, and the advantages of utilizing this control group. Additionally, we have discussed the potential benefits of employing this control group for identifying new therapeutic targets for osteoarthritis (OA). Furthermore, we have further examined the potential mechanisms of the newly identified targets in the results section (line 306-313, page 15), regarding their potential involvement in OA. In the discussion section, we expanded the discussion and possible mechanism of PSD95-NOS-NO's involvement in the chronic inflammatory microenvironment of OA, with the recent relevant evidence, such as the study of Caric et al, Int J Mol Sci, 2021.

  1. Materials and methods

I recommend you to named the first subsection Study design, and begin this section with an information what type of study you have performed, in which time period and where. This subsection should contain the Ethical approval for conducting the study which you have already stated, however please state the date when the Ethical approval was gained. If you register your study on any protocol this information should also be stated here (e.g., ClinicalTrials.gov).

Response: We thank the reviewer for his/her thoughtful consideration.We have named Study design in the first subsection. This study aims to provide a descriptive analysis within an observational research setting.This study was conducted from December 16, 2019, to December 16, 2022, and finally enrolled 10 patients for RNA sequencing in Shanghai Renji hospital, while the cartilage samples of another 5 patients were used for cell cultivation-related experiments. All human experiments in this study adhered to the regulations of the Ethics Com-mittee at Ren Ji Hospital (ethical approval identification number: 2019-166, approved on December 16, 2019).

4.10. Statistical analysis

Should be for some results a more appropriate to show them as median and IQR and then analyze by Wilcoxon test?

How did you arrive to the final sample size. Did you perform any type of sample size calculation?

Response: We thank you for your suggestions.

Regarding the data analysis of the article, we used a small number of samples (n=3 or 6), and the sample distribution conformed to the normal distribution, so we considered using the mean standard deviation and the t-test as more appropriate. Researchers need to collect at least 3 clinical samples from each group, and when the variation is large, at least 5-6 samples from each group. When determining our sample size settings, we also surveyed the sample sizes used by other research groups, such as M.F. Rai, 2019 collected cartilage samples from 10 patients after TKA and 10 patients after APM for transcript differential analysis. Banu Bayram, 2020 collected posterior capsule specimens that underwent TKA due to fibrosis, osteoarthritis, and non-infectious causes, with 4 cases in each group. Referring to the sample size settings of other researchers, we collected 10 cartilage samples after TKA after a series of inclusion and exclusion criteria screening and used a unique self-contrast model.

  1. Results

Please begin this section with the period of conducting your study and with the information of the number if patients ultimately analyzed.

Response: We thank you for your suggestions.

This study was conducted from December 16, 2019, to December 16, 2022, and finally enrolled 10 patients for RNA sequencing. The relevant information is described in the Methods (lines 324-326, page 16) and Results section (lines 79-80, page 2).

  1. Discussion

Discussion section is definitely too short, so try to better explain your Results and compare them and further expand with the findings of recent similar studies.

Please state the limitations of your study at the end of Discussion.

Response: We thank the reviewer for the valuable suggestion.

We have made revisions to the discussion section to emphasize the results of our study and the differences from previous research. We have also listed the limitations of our study. This study has certain limitations. (1) The number of included samples for RNA sequencing in OA cartilage is limited. (2) The underlying mechanisms by which PSD95 promotes OA were not extensively explored. Previous studies have reported that PSD95 can activate the NOS-NO pathways, suggesting its potential involvement in the release of inflammatory factors. In accordance with the reviewer's suggestions, appropriate changes have been made and supplemented in the Results and Discussion sections of the manuscript. (page 15)

References

Please add some relevant recent references, firstly to expand the Discussion section of your work.

Response: According to the reviewer's comments, we have revised and expanded the discussion section and added more references in this manuscript.

Round 2

Reviewer 1 Report

Congratulations to the authors for making the requested suggestions on their manuscript.